# Peripapillary Retinal Vascular Involvement in Early Post-COVID-19 Patients

**DOI:** 10.3390/jcm9092895

**Published:** 2020-09-08

**Authors:** Alfonso Savastano, Emanuele Crincoli, Maria Cristina Savastano, Saad Younis, Gloria Gambini, Umberto De Vico, Grazia Maria Cozzupoli, Carola Culiersi, Stanislao Rizzo

**Affiliations:** 1Ophthalmology Unit, Fondazione Policlinico Universitario A. Gemelli IRCCS, 00196 Rome, Italy; asavastano21@gmail.com (A.S.); emanuelecrincoli1@gmail.com (E.C.); gambini.gloria@gmail.com (G.G.); umbertodevico@gmail.com (U.D.V.); mgcozzupoli@gmail.com (G.M.C.); c.carola@live.it (C.C.); stanislao.rizzo@gmail.com (S.R.); 2Department of Ophthalmology, Catholic University of “Sacro Cuore”, 00168 Rome, Italy; 3Department of Ophthalmology, Western Eye Hospital, Imperial College Healthcare NHS Trust, London NW1 5QH, UK; saad.younis@nhs.net; 4Neuroscience Institute, Consiglio Nazionale delle Ricerche, Istituto di Neuroscienze, 56124 Pisa, Italy

**Keywords:** OCT angiography, peripapillary capillary perfusion, personalized medicine, SARS-CoV-2

## Abstract

The ability of severe acute respiratory syndrome coronavirus 2 (SARS-CoV-2′s) to cause multi-organ ischemia and coronavirus-induced posterior segment eye diseases in mammals gave concern about potential sight-threatening ischemia in post coronavirus disease 2019 patients. The radial peripapillary capillary plexus (RPCP) is a sensitive target due to the important role in the vascular supply of the peripapillary retinal nerve fiber layer (RNFL). Eighty patients one month after SARS-CoV-2 infection and 30 healthy patients were selected to undergo structural OCT (optical coherence tomography) and OCTA (optical coherence tomography angiography) exams. Primary outcome was a difference in RPCP perfusion density (RPCP-PD) and RPCP flow index (RPCP-FI). No significant difference was observed in age, sex, intraocular pressure (IOP) and prevalence of myopia. RPCP-PD was lower in post SARS-CoV-2 patients compared to controls. Within the post-COVID-19 group, patients with systemic arterial hypertension had lower RPCP-FI and age was inversely correlated to both RPCP-FI and RPCP-PD. Patients treated with lopinavir + ritonavir or antiplatelet therapy during admission had lower RPCP-FI and RPCP-PD. RNFL average thickness was linearly correlated to RPCP-FI and RPCP-PD within post-COVID-19 group. Future studies will be needed to address the hypothesis of a microvascular retinal impairment in individuals who recovered from SARS-CoV-2 infection.

## 1. Introduction

Coronaviruses are a subfamily of single-stranded positive-sense RNA viruses widely diffused among animal species. Their genome length (26–30 kb) is one of the largest known sequences among RNA viruses. Four known genera of coronaviruses (Alfacoronaviruses, Betacoronaviruses, Gammacoronaviruses, and Deltacoronaviruses) exist, but only seven species have the ability to infect humans: 229-E, NL-63, OC-43, HK-U1, MERS-CoV, SARS-CoV-1, and SARS-CoV-2 [1]. The earliest reports of endemic human CoV date back to the 1960s, when HCoV-OC43 and -229E were described. HCoV-NL63 and -HKU1 were discovered only in 2004 and 2005, respectively. In addition to these four endemic HCoVs, two epidemic CoVs have emerged in humans in the last two decades, severe acute respiratory syndrome (SARS)-CoV and the Middle East respiratory syndrome (MERS)-CoV discovered in 2003 and followed by 2012 in Saudi Arabia [2]. Indeed, the first coronavirus epidemic, caused by Severe Acute Respiratory Syndrome coronavirus 1 (SARS-CoV-1), affected 8000 people causing 774 death around the world [3,4]. Severe Acute Respiratory Syndrome 2 (SARS-CoV-2), discovered in December 2019 due to the outbreak of an epidemic in Wuhan region (China), is currently causing a pandemic leading to dramatic human and economic losses. By now SARS-CoV-2 has led to the death of 500,000 people globally and this number is constantly growing [5,6,7].

SARS-CoV-2 is responsible for a high morbidity burden in affected patients, because the virus causes permanent damage to organs involved during the acute phase [8]. SARS-CoV-2 RNA was detected, using RT-PCR tests, on conjunctival swabs of 0–15% of infected patients [9,10]. Although some studies described the positivity for the viral genome in tear samples from SARS-CoV-2 patients, the relevance of these findings, in a large cohort of patients, is still controversial [11]. Forty percent of patients with positive conjunctival swabs reported symptomatic conjunctivitis. Occurrence of ocular symptoms, as a first symptom or concurrently with the systemic involvement, were equally frequent [12]. Reported signs and symptoms are: bilateral/unilateral diffuse hyperemia, viscous whitish secretion in the conjunctival sac, foreign body sensation, and excessive tearing [13,14].

To our knowledge, no severe posterior segment involvement has been described in patients after SARS-CoV-2 infection. According to coronavirus manifestations in other mammals, it is reasonable to believe that ocular sequelae might affect post-infectious patients [15]. For example, feline infectious peritonitis virus (FIPV) is an Alphacoronavirus that can affect feline species causing pyogranulomatous anterior uveitis, choroiditis with retinal detachment and retinal vasculitis [15].

Likewise, the neurotropic JHM strain of the mouse hepatitis virus (JHMV) showed involvement of the posterior pole in the eye. JHMV-infected mice were subsequently utilized for intravitreal inoculation to study the mechanisms of virus-induced retinal degeneration. Currently, this model of retinal degeneration, known as the experimental coronavirus retinopathy (ECOR), is used to examine genetic and host immune responses that may contribute to retinal disease [16]. ECOR is a biphasic disease, characterized by inflammation in the early phase and retinal degeneration occurring after viral clearance. The second phase is characterized by progressive loss of photoreceptors and ganglion cells as well as thinning of the neuroretina. Further, MHV-A59 (a different neurotropic strain of MHV) has been used to generate viral-induced optic neuritis models [17].

Our current understanding of the retinal vascular network is based on pioneering work on primate histology [18,19]. Wang et al. [20] recently described two complexes and four plexuses in the macular region: the superficial (SVP), intermediate (IVP), and deep vascular plexus (DVP), as well as the radial peripapillary capillary plexus (RPCP). RPCP is a peculiar vascular plexus running alongside RNFL fibers, playing an important role in the supply of the densely packed nerve fiber bundles in this region [21]. Clinical imaging of the ocular circulation, for decades dominated by fluorescein angiography and indocyanine green angiography, can now concurrently rely on optical coherence tomography angiography (OCTA) [22]. The principle of OCTA is to use the variation in OCT signals caused by moving particles, such as red blood cells (RBC) and other blood elements, as the contrast mechanism for imaging blood flow [23]. OCTA allows a detailed examination of different vascular layers as well as important functional information on the blood supply of the retina and choroid [24].

The purpose of our study was to investigate peripapillary vascular impairment in post SARS-CoV-2 patients compared to controls, analyzing OCTA imaging of the RPCP and structural OCT parameters.

## 2. Materials and Methods

### 2.1. Study Design and Patient’s Selection

This observational case-control study was performed at and supported by the Fondazione Policlinico Universitario A. Gemelli IRCCS and the Università Cattolica del Sacro Cuore, Rome, Italy. The study was designed by the investigators of “Gemelli Against COVID Post-Acute Care Study Group” [25].

Two groups of age matched consecutive patients were selected from the hospitals’ patient databases. The post-COVID-19 group included 80 patients who contracted and successively recovered from SARS-CoV-2 infection from 1 March 2020 to 1 June 2020. All clinical data concerning the SARS-CoV-2 infection diagnosis and clinical course were obtained consulting patients records. SARS-CoV-2 infection was testified by two successive oropharyngeal swabs positive for the SARS-CoV-2 genome. Recovery was testified by the simultaneous presence of two consecutive negative swabs, resolution of symptoms (if any), and detection of anti-SARS-CoV-2 IgGs in blood samples.

The CONTROL group consisted of 30 patients without previous or current SARS-CoV-2 infection, which were randomly selected from the hospital databases. Inclusion criteria were: two successive oropharyngeal swabs resulted negative for the SARS-CoV-2 genome, absence of symptoms suggestive of SARS-CoV-2 infection during the previous months, and blood detection of anti-SARS-CoV-2 IgGs resulted negative.

Exclusion criteria for both groups were: choroidal atrophy, high myopia, exudative AMD, previous episode of central serous chorioretinopathy, glaucoma, acquired and hereditary optic neuropathy, hereditary retinal diseases, demyelinating disorders, neurodegenerative disorders, and keratoconus. Furthermore, image quality was mandatory and was defined acceptable if > 7/10.

The study adhered to the declaration of Helsinki. Ethical Committee authorization (ID number: 003220/20) was obtained prior to the study. A complete explanation of the study protocol was fully provided and informed consent was collected for all study participants.

### 2.2. Procedures and Instruments

Patients who adhered to the study underwent a clinical and instrumental evaluation. Patients from the post-COVID-19 group were recruited at 1 month from hospital discharge.

Each patient underwent a comprehensive ophthalmologic examination, including best corrected visual acuity, slit lamp anterior segment observation (SL9900 Slit Lamp, CSO, Florence, Italy), IOP measurement (Goldman tonometry) and dilated fundus inspection and photography (Cobra HD Fundus Camera, CSO, Florence, Italy).

Structural OCT and OCTA analysis were performed by an expert physician using Spectral Domain Zeiss Cirrus 5000-HD-OCT Angioplex (sw version 10.0, Carl Zeiss, Meditec, Inc., Dublin, CA, USA). One eye for each patient was chosen randomly to undergo the examination. In the case of unilateral eye disease, the other eye was selected. Group attribution was blinded to the examiner performing the OCT.

Structural OCT images consisted of the Optic Disc Cube 200 × 200, and Macular Cube 512 × 128 patterns. The subfoveal choroidal thickness (SCT) was manually measured on cross-sectional OCT B-scans [26]. Two independent masked graders individually assessed all choroidal thickness measurement in the fovea region, from the rear edge of the RPE to the choroid-sclera junction. The agreement between the two observers was determined through Bland–Altman plot.

OCTA scan protocol was 4.5 × 4.5 mm centered into the disc in healthy and post-COVID-19 patient’s eyes. Two-dimensional en face OCT angiograms of the RPC layer were generated with automated segmentation software (Cirrus 10.0), with the RPC defined as the segment extending superficially from the inner limiting membrane to the posterior surface of the RNFL. En face images were processed using custom software with an interactive interface [26]. The software used a method combining a global threshold, Hessian filter, and adaptive threshold to generate binary vessel maps, which were used to calculate quantitative indices of blood flow in MATLAB (R2017a; MathWorks, Inc., Natick, MA, USA). The peripapillary flow index was defined as the average decorrelation value in the peripapillary region of the en face retinal angiogram. The peripapillary vessel density was defined as the proportion of the total area occupied by vessels. The blood vessels were defined as the pixels with decorrelation values over the threshold in the noise region, which were two standard deviations higher than the mean decorrelation value. The avascular zone of the ONH was manually selected to establish baseline background noise level for global thresholding, and the ONH was excluded from quantification [27]. Finally, RPCP perfusion density (RPCP-PD) and RPCP flow index (RPCP-FI) were collected and used for analysis.

### 2.3. Outcome Measures and Analyzed Confounders

The primary endpoint was a difference in the RPCP-FI and RPCP-PI. The following parameters were chosen as secondary outcome measures: GCC average thickness, RNFL average thickness, disc area, CD ratio, central foveal thickness, choroidal thickness. Furthermore, we performed an additional analysis within the post-COVID-19 group correlating the primary outcome measures with the other examined variables to detect potential risk factors for RPCP impairment in post SARS-CoV-2 patients.

Potential confounders taken into account were: age, systemic autoimmune and inflammatory diseases, axial myopia > 1D (dichotomous variable) [28], systemic arterial hypertension, and diabetes.

### 2.4. Statistical Analysis

The sample size calculation was performed using G*power (3.1.9.7 software, Düsseldorf, Germany). Statistical analysis was conducted using SPSS software (IBM SPSS Statistics 26.0, Ontario, Canada). Alpha and beta error were established at 5% and 20%, respectively.

The following variables were considered as continuous quantitative variables: age, central foveal thickness, choroidal thickness, GCC average thickness, RNFL average thickness, disc area, CD ratio, RCP perfusion density, and RCP flow index. Assimilability to normal distribution was evaluated using Shapiro–Wilk test. Univariate comparison between the two groups was performed using T-test for independent groups. Linear correlations between quantitative variables were performed using Spearman’s Test. The remaining variables were considered as qualitative variables. The univariate comparison between the groups was performed by means of a Chi^2^ test. Logistic regression analysis was performed to evaluate the actual strength of the associations detected by the univariate analysis. A Bonferroni corrected *p* value < 0.01 was considered to establish the statistical significance of the results.

## 3. Results

Demographic and anamnestic data were collected by the same physician performing the visit and are reported in Table 1. Results from the descriptive analysis in post-COVID-19 group are summarized in Table 2. The mean age in the group was 52.9 ± 13.5 years with 57.5% male patients. The prevalence of systemic arterial hypertension, diabetes, and autoimmune or inflammatory systemic diseases was 23.8%, 42.5%, and 23.8%, respectively. Almost 14% of the patients presented ≥ 1D of axial myopia. Mean IOP at the visit was 16.2 ± 1.5 mmHg.

Data collected from the hospital admission for SARS-CoV-2 infection revealed that only 6.25% of the patients spent part of their recovery in the intensive care unit (ICU) and 8.8% required the support of noninvasive ventilation (NIV) during the hospital stay. The mean duration of the ICU recovery was 3.2 days. Medical therapy was administered as follows: 68.8% were treated with hydroxychloroquine, 33.8% with lopinavir + ritonavir, 43.8% with darunavir + ritonavir, 41.3% with anticoagulant therapy (heparin), and 35% with azithromycin. Other drugs used for systemic support were antiplatelet therapy (aspirin or clopidogrel) in 7.5% of the patients and corticosteroids in 5% of the patients. Finally, 45% of the post-COVID-19 group reported tearing, dry eye, or red eye during the infectious period. In the CONTROL GROUP, mean age was 48.5 ± 13.4 years, and 43.3% (13/30) were males and 56.6% (17/30) females. Few patients (10%) reported systemic arterial hypertension while none of them was affected by diabetes or systemic autoimmune or inflammatory diseases. Axial myopia ≥ 1D was present in 13.3% of the subjects. Mean IOP was 14.4 ± 2.1 mmHg.

No statistically significant differences between the 2 groups were detected in terms of age, gender, IOP at the visit or prevalence of axial myopia ≥ 1D. The post-COVID-19 group showed a higher prevalence of systemic arterial hypertension (*p* < 0.03), diabetes (*p* < 0.001) and autoimmune or inflammatory systemic diseases (*p* < 0.001).

One of the most significant differences between the two groups was observed in the RPCP-PD analysis. (Figure 1) Indeed, lower RPCP-PD value in post-COVID-19 group compared to the control group (*p* < 0.04) was observed (Figure 2).

This difference was further confirmed by the binary logistic regression analysis including all potential confounders (*p* < 0.039). None of the other outcome measures showed statistically significant differences between the two groups (Table 3). Bland–Altman analysis of subfoveal choroidal thickness measurement revealed a good agreement between graders (Bias = 1.23, CI = 0.94–1.41, LA = 28.3%).

Within the post-COVID-19 group, patients affected by systemic arterial hypertension were characterized by a statistically relevant reduction of the RPCP-FI (*p* < 0.001). Moreover, age distribution showed an inverse linear correlation with both RPCP-FI (*p* < 0.001) and RPCP-PD (p < 0.01). Furthermore, patients treated with lopinavir + ritonavir during SARS-CoV-2 infection showed both a lower RPCP-FI (*p* < 0.01) and a lower RPCP-PD (*p* < 0.01) compared to the other patients in the POST-COVID-19 group. A similar result was demonstrated in patients treated with antiplatelet therapy during hospital recovery. Indeed, in these patients, RPCP-FI and RPCP-PD were statistically lower than those not treated (respectively *p* = 0.004 and *p* = 0.003). A detailed description of the within post-COVID-19 group analysis is available in Table 4.

Spearman’s Test revealed a statistically significant linear correlation between RNFL average thickness and both RPCP perfusion density (*p* < 0.001) (Figure 3) and RPCP flow index (*p* < 0.001) (Figure 4) within the post-COVID-19 group. Contrarily, SCT showed no significant linear correlation with RPCP parameters (Table 5).

## 4. Discussion

COVID-19 caused by SARS-CoV-2 evolved into a severe pandemic moving the whole of humanity into jeopardy. While the main manifestation has been observed in the respiratory tract, multi-systemic organ involvement has been observed. According to our results, post-COVID-19 patients have a lower RPCP PD and a normal RPCP FI compared to the general population. These findings suggest an impairment in the blood supply to the peripapillary RNFL in patients who recovered from SARS-CoV-2 infection. It is, to our knowledge, the first published study to detect this potential threat. Moreover, RPCP microvascular impairment is more evident in older patients (RPCP PD and RPCP FI are both inversely correlated with age in post-COVID-19 group) and patients affected by systemic arterial hypertension (lower RPCP FI compared to the general population). In addition, patients treated with antiplatelet therapy or lopinavir + ritonavir during admission are more susceptible to RPCP impairment after SARS-CoV-2 infection. In addition, both RPCP PD and RPCP FI are linearly correlated to average RNFL thickness in early post SARS-CoV-2 patients. None of the structural OCT parameters proved to be significantly different between the study groups.

Recent research has focused on SARS-CoV-2′s ability to damage the vascular endothelium causing irreversible ischemic damage to multiple organs; this microcirculatory impairment leading to functional disorders in all inner organs is believed to be the ultimate cause for the high mortality and morbidity rate [29]. Indeed, macro- and microvascular thrombotic processes in severe SARS-CoV-2 infection cases cause a high burden of complications [30,31]. Several elements contribute to endothelial disruption during SARS-CoV-2 infection, such as complement activation, hypoxia, platelets, and thyroxin kinases [32]. Endothelial dysfunction together with a generalized inflammatory state and complement elements may contribute to the overall pro-coagulative state described in COVID-19 patients, leading to occlusions in veins and arteries [33]. Due to this phenomenon, COVID-19 has been shown to cause rare clinical events such as atypical thromboses (renal veins, uterine veins, and mesenteric vessels) and myocardial micro-thrombotic vessels. Endothelial derangement and increased permeability are also reported to be early hallmarks of organ damage in patients with COVID-19 [32]. In this panorama, our study investigated the involvement of the retinal capillary microcirculation focusing on the radial peripapillary capillary plexus, which is considered to be crucial for the homeostasis and function of the retinal ganglion cells and their axons. RPCP density is highly correlated to RNFL thickness and visual field index in glaucoma patients [33,34]. Moreover, a reduction in RPCP density has been demonstrated to be an early sign of glaucoma [35,36]. RPCP density and flow index reduction are also correlated to visual acuity and visual field loss in non-arteritic ischemic optic neuropathy [37,38]. On the contrary, a RPCP flow and density impairment can be the consequence of retinal neural remodeling secondary to optic nerve axonal degeneration [39]. Our study examined this aspect outlining the correlation of the RPCP perfusion density and RPCP flow index with the RNFL average thickness also in early post-COVID-19 patients.

In our opinion, the conflicting results deriving from the comparison of post-COVID-19 patients with the healthy controls on the field of RPCP integrity could be attributable to the characteristics of the examined groups. First, our shortage of healthy control patients led to an asymmetry in the sample sizes of the two groups. Furthermore, our patients recruited in the post-COVID-19 group manifested a mild to moderate variant of the infection: only 6.25% of the subjects required admission to the ICU department and the cases of pulmonary and venous thromboembolism were only 4 in 80 patients. In this perspective, our results are in agreement with those of Mazzaccaro et al. [40], as we analyzed a cohort of COVID-19 patients with mild disease progression. Moreover, within the post-COVID-19 group, both RPCP-FI and RPCP-PD are linearly inversely correlated with age. In addition, patients affected by systemic arterial hypertension showed a statistically lower RPCP-FI compared to other patients in the post-COVID-19 group. In this regard, it is interesting to notice that patients in the post-COVID-19 group showed a lower mean age, a lower prevalence of diabetes and systemic arterial hypertension, and a higher prevalence of females (typically affected by milder manifestations of the disease) compared to the reported SARS-CoV-2 epidemiologic data [38]. These data additionally confirm our hypothesis of an altered group composition being responsible for a mild significance of the result.

Curiously, our results show that patients treated with antiplatelet therapy during hospitalization were characterized by lower RPCP FI and lower RPCP PD. The role of platelets in inducing or amplifying the endothelial damage in COVID-19 patients is still a matter of discussion. A low platelet count, possibly due to destruction, bone marrow infection, or autoimmune phenomena, was reported to cause a five-fold mortality rate increase in COVID-19 patients and the rates reported were very heterogeneous among the analyzed studies [41,42]. However, the opposite is more common in COVID-19 patients: usually the platelet count is higher than in patients with sepsis or ARDS. Increased serum levels of thrombopoietin caused by pulmonary inflammation have been supposed to explain this phenomenon [43,44]. We hypothesize that this finding in our study could be due to the administration of adjunctive drugs in patients with more severe clinical condition, causing more systemic microvascular damage.

This occurrence could possibly explain another unexpected finding of our study: the use of Lopinavir + ritonavir during recovery was associated with lower RPCP-FI and RPCP-PD. Another possible explanation can be related to antiviral drug that may induce endothelial damage. Endothelial damage, secondary to medications, is reported in the literature for several substances: the damage caused by ponatinib, for example, is mediated by NOTCH1 hyperactivation, but also propranolol and sirolimus inhibit endothelial proliferation [45]. Similarly, carteolol induces apoptosis in corneal endothelial cells by caspase- and mitochondria-dependent pathways [46]. ACE2, a SARS-CoV-2 target, inhibits proliferation of endothelial cells; however, it also reduces endothelial inflammation [47]. Finally, steroids induce apoptosis in bone endothelial cells causing osteonecrosis, but this effect has not been proven in retinal capillary cells [48]. Nowadays, no report of lopinavir + ritonavir induced retinal endothelial damage has been described.

In conclusion, it is important to highlight how differential analysis of risk factors for microvascular peripapillary involvement in post-COVID-19 infection represents a valuable tool for personalized medicine.

Despite being to our knowledge the first study to address a potential RPCP impairment in patients who recovered from SARS-CoV-2 infection, the results are undermined by some limitations. First, the selected sample of post-COVID-19 patients is not fully representative of the average post-COVID-19 population of patients. Moreover, a larger cohort of healthy controls would be needed to increase the power of the study. Future studies will be needed to address the question of a potential difference in RPCP perfusion between healthy subjects and individuals who recovered from SARS-CoV-2 infection. Likewise, another future prospective will be to investigate whether peripapillary vascular damage can be a reversible occurrence in these patients.

## 5. Conclusions

OCT angiography provided several information of RPCP circulation. RPCP-PD was lower in post SARS-CoV-2 patients compared to controls. Patients treated with lopinavir + ritonavir or antiplatelet therapy during admission had lower RPCP-FI and RPCP-PD. Within the post-COVID-19 group, patients with systemic arterial hypertension had lower RPCP-FI and age was inversely correlated to both RPCP-FI and RPCP-PD. RNFL average thickness was correlated to RPCP-FI and RPCP-PD within post-COVID-19 group. Future studies will be needed to confirm our hypothesis of a microvascular retinal impairment in individuals who early recovered from SARS-CoV-2 infection.

## Figures and Tables

**Figure 1 jcm-09-02895-f001:**
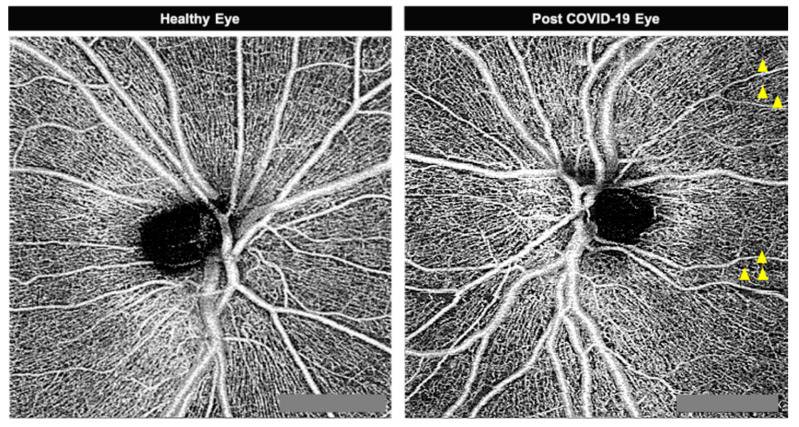
Optical Coherence tomographic angiography (4.5 × 4.5 mm) centered into the disc in healthy and post COVID-19 patient’s eyes. The yellow arrowhead shows the perfusion defect.

**Figure 2 jcm-09-02895-f002:**
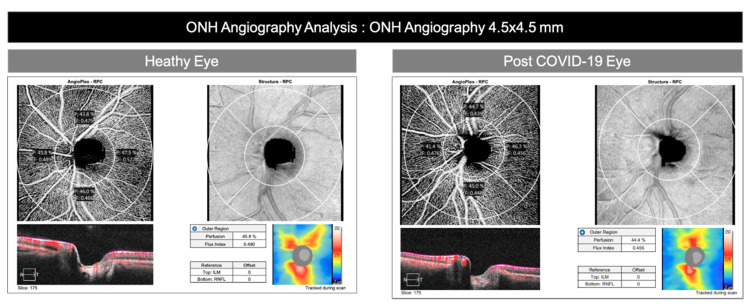
Healthy and post-COVID-19 eyes. Angioanalysis of optic nerve head analysis (ONH = optic nerve head Angiography, 4.5 × 4.5 mm). Lower RPCP-PD value in post-COVID-19 eyes compared to the healthy group (*p* < 0.04) was observed.

**Figure 3 jcm-09-02895-f003:**
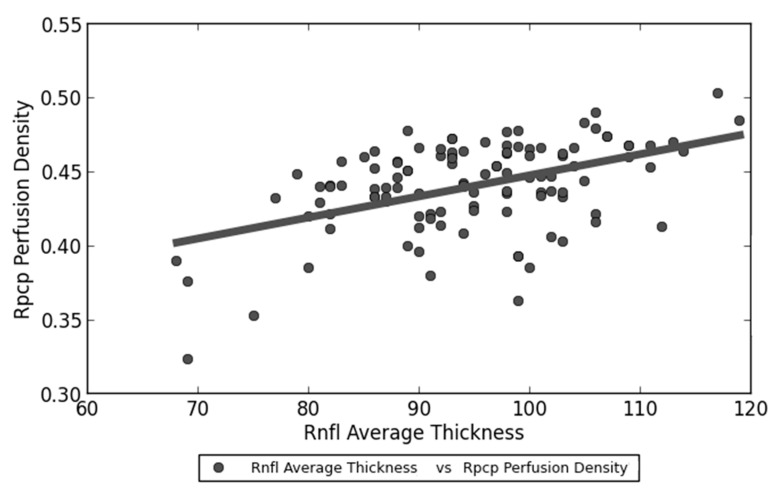
Linear correlation between RNFL (Retinal nerve fiber layer) average thickness and RPCP (Radial peripapillary capillary plexus) perfusion density within post-COVID-19 group patients.

**Figure 4 jcm-09-02895-f004:**
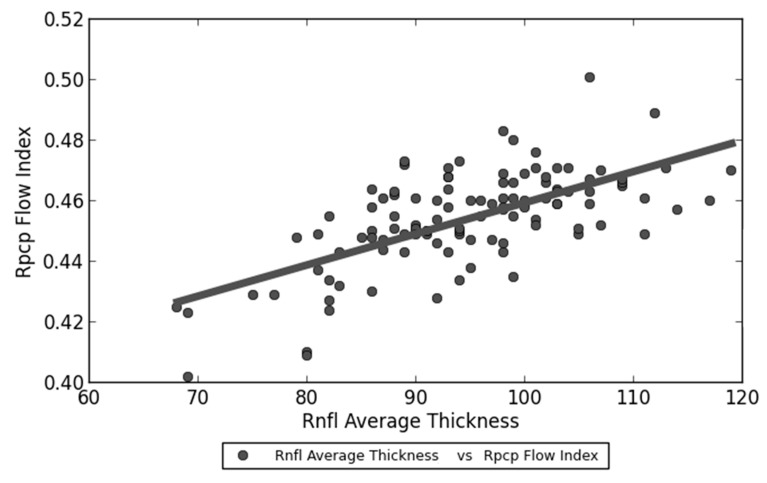
Linear correlation between RNFL (Retinal nerve fiber layer) average thickness and RPCP (Radial peripapillary capillary plexus) flow index within post-COVID-19 group patients.

**Table 1 jcm-09-02895-t001:** Demographic and anamnestic data were collected by the same physician performing the visit.

Investigated Anamnestic Data	
Post-COVID-19 Group	Control Group (Healthy Patients)
Age	Age
Sex	Sex
Date of birth	Date of birth
Height (cm)	Height (cm)
Weight (Kg)	Weight (Kg)
Sanitary code	Sanitary code
Preexisting systemic diseases	Preexisting systemic diseases
Preexisting ocular diseases	Preexisting ocular diseases
Familiar diseases	Familiar diseases
Date and findings of the oropharyngeal swabs	Date and findings of the oropharyngeal swabs
Results from the serologic exam	Results from the serologic exam
Reported ocular and systemic symptoms	Reported ocular and systemic symptoms
Duration of hospital stay	
Hosting hospital department	
Administered treatment	
Supportive treatment	
Complications occurred during hospital stay	

**Table 2 jcm-09-02895-t002:** Descriptive analysis of the study groups. IOP values are the mean of three measurements performed at the time of the study examination. Drugs are intended as administered during hospitalization.

Variable	Post-COVID-19	Controls	*p*
Age (years)	52.9 ± 13.5	48.5 ± 13.4	0.71
Sex	M = 46/80 (57.5%) F = 34/80 (42.5%)	M = 13/30 (43.3%) F = 17/30 (56.6%)	0.26
Systemic arterial hypertension	19/80 (23.8%)	3/30 (10%)	0.03
Diabetes	34/80 (42.5%)	0/30 (0%)	<0.001
Autoimmune or inflammatory diseases	19/80 (23.8%)	0/30 (0%)	<0.001
Myopia > 1D	11/80 (13.8%)	(13.3%)	0.87
IOP	16.2 ± 1.5 mmHg	14.4 ± 2.1 mmHg	0.34
Red/dry eye during infection	36/80 (45%)		
Days since symptoms onset	60.3 ± 13.6		
Days since hospital discharge	36.1 ± 12.9		
ICU admission	5/80 (6.25%)		
Oxygen therapy	33/80 (41.25%)		
Noninvasive ventilation	7/80 (8.8%)		
Pulmonary embolism	2/80 (2.5%)		
Venous thrombosis	2/80 (2.5%)		
Hydroxychloroquine	55/80 (68.8%)		
Lopinavir + ritonavir	27/80 (33.8%)		
Darunavir + ritonavir	35/80 (43.8%)		
Azithromycin	28/80 (35%)		
Heparin	33/80 (41.3%)		
Antiplatelet therapy	6/80 (7.5%)		
Corticosteroids	4/80 (5%)		

IOP = Intraocular pressure; ICU = Intensive care unit.

**Table 3 jcm-09-02895-t003:** Analysis of primary and secondary outcomes in the study groups.

Outcomes	Variable	Post-COVID-19 Mean ± SD (CI)	Control Mean ± SD (CI)	*p* *t* Test
Primary Outcomes	RPCP flow index	0.454 ± 0.017 (0.450–0.457)	0.456 ± 0.012 (0.452–0.461)	0.42
	RPCP perfusion density	0.437 ±0.031 (0.430–0.444)	0.450 ± 0.025 (0.441–0.459)	0.041
Secondary Outcomes	RNFL average thickness	94.09 ± 10.77 (91.77–96.43)	96.50 ± 7.78 (93.72–99.28)	0.26
	GCC average thickness	81.21 ± 8.67 (79.33–83.09)	80.87 ± 5.80 (78.791–82.94)	0.84
	CD ratio	0.43 ± 0.18 (0.39–0.47)	0.43 ± 0.14 (0.38–0.48)	0.90
	Disc area	1.76 ± 0.33 (1.69–1.83)	1.688 ± 0.36 (1.56–1.82)	0.31
	Central foveal thickness	263.83 ± 24.28 (258.57–269.08)	260.1± 22.6 (252.0–268.2)	0.46
	Subfoveal choroidal thickness	310.463 ± 81.60 (292.80–328.13)	293.5 ± 86.56 (262.52–324.48)	0.34

RPCP = radial peripapillary capillary plexus; RNFL = Retinal nerve fiber layer; GCC = ganglion cell complex; CD = Cup disc.

**Table 4 jcm-09-02895-t004:** Analysis of the variations of the primary outcome measures within the post-COVID-19 according to the other examined variables. The drugs and clinical complications occurred during the hospital recovery are listed.

Variable	RPCP Flow Index Means ± SD (CI)	*p*	RPCP Perfusion Index Means ± SD (CI)	*p*
Age	R = −0.421 Slope = −340.216 Intercept = 207.199	**<0.001**	R = −0.278 Slope = −145.568 Intercept = 116.528	**0.01**
Sex	M = 0.453 ± 0.02 (0.448–0.458) F = 0.454 ± 0.02 (0.449–0.460)	0.70	M = 0.435 ± 0.03 (0.425–0.444) F = 0.441 ± 0.03 (0.432–0.450)	0.39
Systemic arterial hypertension	Absent = 0.457 ± 0.01 (0.453−0.461) Present = 0.442 ± 0.02 (0.433–0.451)	**<0.001**	Absent = 0.439 ± 0.03 (0.432–0.446) Present = 0.431 ± 0.04 (0.413–0.449)	0.31
Systemic autoimmune or inflammatory diseases	Absent = 0.453 ± 0.02 (0.449–0.457) Present = 0.455 ± 0.02 (0.447–0.463)	0.68	Absent = 0.437± 0.03 (0.429–0.444) Present = 0.439 ± 0.03 (0.425–0.453)	0.75
Diabetes	Absent = 0.453 ± 0.02 (0.448–0.458) Present = 0.454 ± 0.02 (0.448–0.459)	0.91	Absent = 0.433 ± 0.03 (0.424–0.442) Present = 0.442 ± 0.03 (0.432–0.453)	0.17
Axial myopia > 1D	Absent = 0.453 ± 0.02 (0.449–0.458) Present = 0.454 ± 0.01 (0.447–0.462)	0.87	Absent = 0.438 ± 0.03 (0.430–0.446) Present = 0.434 ± 0.02 (0.422–0.446)	0.68
IOP in study examination	R = 1.28 Slope = 41.712 Intercept = 10.553	0.53	R = 2.83 Slope = 57.931 Intercept = 4.006	0.61
Red/dry eye during infection	Absent = 0.453 ± 0.02 (0.448–0.459) Present = 0.454 ± 0.02 (0.449–0.459)	0.87	Absent = 0.436 ± 0.03 (0.427–0.446) Present = 0.438 ± 0.03 (0.429–0.448)	0.78
Hydroxychloroquine	No = 0.455 ± 0.02 (0.449–0.462) Yes = 0.453 ± 0.02 (0.448–0.457)	0.57	No = 0.442 ± 0.03 (0.431–0.453) Yes = 0.435 ± 0.03 (0.427–0.444)	0.36
Lopinavir + ritonavir	No = 0.457 ± 0.01 (0.453–0.461) Yes = 0.447 ± 0.02 (0.439–0.455)	**0.01**	No = 0.443 ± 0.03 (0.435–0.450) Yes = 0.425 ± 0.03 (0.413–0.438)	**0.01**
Darunavir + ritonavir	No = 0.451 ± 0.02 (0.445–0.457) Yes = 0.457 ± 0.01(0.453–0.461)	0.10	No = 0.433 ± 0.03 (0.424–0.443) Yes = 0.442 ± 0.03 (0.433–0.452)	0.19
Heparin	No = 0.454 ± 0.02 (0.450–0.458) Yes = 0.453 ± 0.020(0.445–0.460)	0.71	No = 0.438 ± 0.03 (0.430–0.446) Yes = 0.435 ± 0.03 (0.422–0.448)	0.68
Antiplatelet therapy	No = 0.455 ± 0.02 (0.451–0.458) Yes = 0.43 ± 0.02 (0.409–0.450)	**0.0036**	No = 0.439 ± 0.03 (0.433–0.445) Yes = 0.404 ± 0.06 (0.343–0.464)	**0.0026**
Corticosteroids	No = 0.454 ± 0.02 (0.450–0.457) Yes = 0.453 ± 0.03 (0.419–0.486)	0.89	No = 0.438 ± 0.03(0.431–0.444) Yes = 0.425 ± 0.05 (0.374–0.476)	0.43
ICU admission	No = 0.454 ± 0.02 (0.450–0.458) 0.02 Yes = 0.449 ± 0.01(0.438–0.461)	0.57	No = 0.437 ± 0.03 (0.430–0.444) Yes = 0.44 ± 0.02 (0.424–0.456)	0.84
NIV treatment	No = 0.453 ± 0.02 (0.449–0.457) Yes = 0.461 ± 0.02 (0.444–0.479)	0.22	No = 0.438 ± 0.03 (0.431–0.445) Yes = 0.43 ± 0.02 (0.413–0.448)	0.55
Pulmonary embolism	No = 0.453 ± 0.02 (0.449–0.457) Yes = 0.47 ± 0.01 (0.450–0.490)	0.16	No = 0.438 ± 0.03 (0.431–0.445) Yes = 0.414 0.372–0.457 0.030	0.3
Venous thrombosis	No = 0.453 ± 0.02 (0.449–0.457) Yes = 0.476 ± 0.005 (0.470–0.483)	0.054	No = 0.438 ± 0.03 (0.431–0.445) Yes = 0.417 ± 0.03 (0.369–0.466)	0.37

RPCP = radial peripapillary capillary plexus; IOP = Intraocular pressure; ICU = Intensive care unit; NIV = noninvasive ventilation.

**Table 5 jcm-09-02895-t005:** Spearman’s Test linear correlation between continuous variables of interest.

Linear Correlation Variables (Spearman’s Test)	Result	*p*
RPCP Flow Index—RNFL Average Thickness	R = 0.633 Slope = 429.802 Intercept = −100.844	**<0.001**
RPCP Perfusion Density—RNFL Average Thickness	R = 0.366 Slope = 169.12 Intercept = 20.163	**<0.001**
RPCP Flow Index—Subfoveal Choroidal Thickness	R = 0.238 Slope = 1289.115 Intercept = −274.229	0.031
RPCP Perfusion Density—Subfoveal Choroidal Thickness	R = 0.105 Slope: 353.021 Intercept: 156.133	0.34

RPCP = radial peripapillary capillary plexus; RNFL = Retinal nerve fiber layer.

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
