# Peer review of "Peripapillary Retinal Vascular Involvement in Early Post-COVID-19 Patients"

_jcm, 2020, doi:10.3390/jcm9092895_

Round 1
Reviewer 1 Report
The paper investigated the retinal peripapillary vasculature assessed with OCTA in healthy controls and post COVID 19 patients. Two OCTA based metrics of capillary perfusion capillary density and flow index were assessed. The study topic is very interesting and potentially insightful. Here a few concerns
Page 3- Agreement between the two graders of the choroidal thickness were performed using a Cohen’s Kappa Statistics. Given that the choroidal thickness measures is a continues variable, it is unclear how a Cohen’s Kappa was selected for this comparison. A more suitable statistical tool that assesses continues variables will be appropriate. E.g. Bland and Altman analysis reporting the mean difference and limits of agreement or an ICC.
Page 4- A more thorough description of the quantifying metrics will be helpful. What exactly does the redial peripapillary capillary plexus (RPCP) perfusion density mean? Was this a vessel area density metric or a skeletonized vessel area density. Clearly defining the RPCP-PD and RPCP-FI and how they were computed will prevent ambiguity for comparing the results with future studies. Was there any threshold applied or filter used to enhance the capillary detection for quantification?
Page 4 - was axial myopia compensated as continues variable or as binary variable, and how was the cutoff of 1D selected. Have other potential cutoffs been investigated?
Page 6 - The figure 1 showing an example of the OCTA image of the peripapillary region of the healthy control and post COVID-19 makes it difficult to visually assess and validate the significant difference found in the study. The figure from the post COVID 19 eye clearly has imaging artifacts especially the signal attenuation artifact (in the upper right quadrant) and vessel blurring artifacts. Is this the best image acquired among all 80 eyes post COVID-19? What was the criteria for selecting these two images for showcase? Were the subjects whose images are shown matched for age/ gender/medical conditions or anything? Two high quality OCTA images would be more helpful for comparison.
Author Response
Ophthalmology Unit, Fondazione Policlinico Universitario A. Gemelli IRCCS, Rome, Italy
Catholic University of “Sacro Cuore”
Largo Agostino Gemelli 8
00168 Rome Italy
Director. Prof. Stanislao Rizzo
Corresponding Author:
Maria Cristina Savastano, MD, PhD, Professor
Telephone: +39 338/4443002
E mail: mariacristina.savastano@gmail.com
Roma, Italy, 29/08/2020
Dear Editor,
We are very grateful for the review of our article entitled “Peripapillary Retinal Vascular Involvement in Early Post-COVID-19 Patients”, Manuscript ID: jcm-907580.
The points raised by the Reviewers have provided valuable insights to refine the contents and analysis of our study. We would like to thank the Reviewers for their constructive comments and we try to address the issues providing a point-by-point response to each of their suggestions below.
Reviewer 1
- The paper investigated the retinal peripapillary vasculature assessed with OCTA in healthy controls and post COVID 19 patients. Two OCTA based metrics of capillary perfusion capillary density and flow index were assessed. The study topic is very interesting and potentially insightful. Here a few concerns.
Thank you for your positive feedback. We appreciated it very much.
- Page 3- Agreement between the two graders of the choroidal thickness were performed using a Cohen’s Kappa Statistics. Given that the choroidal thickness measures is a continues variable, it is unclear how a Cohen’s Kappa was selected for this comparison. A more suitable statistical tool that assesses continues variables will be appropriate. E.g. Bland and Altman analysis reporting the mean difference and limits of agreement or an ICC.
We thank the reviewer for this comment. As suggested, we repeated the evaluation of the agreement between the two graders using Bland Altman plot analysis. The modification can be found at page 4 line 6 (“Bland Altman plot analysis"). The results of the analysis are described at page 7 line 15-17.
“Bland Altman analysis of subfoveal choroidal thickness measurement revealed a good agreement between graders (Bias = 1.23, CI = 0.94-1.41, LA = 28.3%).”
- Page 4- A more thorough description of the quantifying metrics will be helpful. What exactly does the redial peripapillary capillary plexus (RPCP) perfusion density mean? Was this a vessel area density metric or a skeletonized vessel area density. Clearly defining the RPCP-PD and RPCP-FI and how they were computed will prevent ambiguity for comparing the results with future studies. Was there any threshold applied or filter used to enhance the capillary detection for quantification?
We agree with the reviewer and implemented the description of the quantifying metrics for RPCP PD and RPCP FI. This change improves the reproducibility of the results for future studies. The implementations can be consulted at page 4 line 15-19 and state as follows:
“The peripapillary flow index was defined as the average decorrelation value in the peripapillary region of the en face retinal angiogram. The peripapillary vessel density was defined as the proportion of the total area occupied by vessels. The blood vessels were defined as the pixels with decorrelation values over the threshold in the noise region, which were two standard deviations higher than the mean decorrelation value. Both assessments can be registered as default in OCTA device used”.
We add also in this section (Response to reviewer) the picture downloaded from the device.
- Page 4 - was axial myopia compensated as continues variable or as binary variable, and how was the cutoff of 1D selected. Have other potential cutoffs been investigated?
Axial myopia influence on RNFL thickness and vascularization is directly proportional to the degree of refractive error and axial length (1). We considered high myopia (> 6 D) to be a profound bias in the analysis, thus we included it among the exclusion criteria. Due to the relatively small amount of myopic patients in the study group, we decided to consider myopia as a qualitative dicotomous variable. The cut off of 1 D was chosen because it is the minimum amount of refractive error that has been proven to determine an influence on RNFL thickness and RPCP function. We added the following terms to the manuscript as a reply to the comment: “(dicotomous variable)” page 4 line 32
We also added the following reference:
Li Y, Miara H, Ouyang P, Jiang B. The Comparison of Regional RNFL and Fundus Vasculature by OCTA in Chinese Myopia Population. J Ophthalmol. 2018;2018:3490962. [Ref. 28]
- Page 6 - The figure 1 showing an example of the OCTA image of the peripapillary region of the healthy control and post COVID-19 makes it difficult to visually assess and validate the significant difference found in the study. The figure from the post COVID 19 eye clearly has imaging artifacts especially the signal attenuation artifact (in the upper right quadrant) and vessel blurring artifacts. Is this the best image acquired among all 80 eyes post COVID-19? What was the criteria for selecting these two images for showcase? Were the subjects whose images are shown matched for age/ gender/medical conditions or anything? Two high quality OCTA images would be more helpful for comparison.
We agree, and added better quality images of heathy and Post COVID-19 eyes. We introduced the yellow arrowhead showing the area of perfusion defect. The information was explained in figure 1 legend.
Figure 1. OCT angiography (4.5x4.5 mm) centered into the disc in healthy and POST-COVID-19 patient’s eyes. The yellow arrowhead shows the perfusion defect.
Reviewer 2
- I contacted the editor regarding this manuscript. Although the paper seems well-written and is of high interest during this awful pandemic, the downloadable manuscript that was available to review DID NOT include any figures/graphs or tables.
We are sorry for the inconvenience and listed the following changes in the manuscript. All the figures have been integrated with explanation in the text.
Lastly, we made 4 additional modifications to the manuscript:
Table 2 has been implemented with the descriptive analysis of the control group and the comparison of the basal characteristics of the groups. Please see attached file named “UPDATED VERSION OF TABLE 2_RPCP COVID”
Figure 3 has been reedited in gray scale version
Figure 4 has been added. It is a graphic of Spearman analysis demonstrating linear correlation between RNFL thickness and RPCP flow index in post COVID 19 patients.
We modified the sentence at page 7 line 2-3 as follows: “Indeed, lower RPCP-PD value in POST COVID-19 group compared to the CONTROL group (p<0.04) has been observed (Figure 2)”
As a consequence of point 3) and 4) the manuscript has been modified as follows: “RPCP perfusion density (p<0.001) (Figure 3) and RPCP flow index (p< 0.001) (Figure 4) within the POST-COVID-19group. Contrary, SCT showed no significant linear correlation with RPCP parameters (Table 5)” (page 10 line 3-5).
Please see attached the file named “Updated graphics” for Figure 3 and Figure 4 consultation.
- My only substantive comment is with respect to Figure 2. The figure is well-designed, but ALL of the labels throughout the figure are WAY TOO SMALL for interpretation by the reader. Please make a full-faith effort to improve the ease of viewing of Figure 2.
We thank the reviewer and decided to reformat the Figure 2 including only the ONH Angiography 4.5x4.5 mm details as follows:
Figure 2. Healthy and POST-COVID-19 eyes. Angioanalysis of optic nerve head analysis (ONH Angiography 4.5x4.5 mm). Lower RPCP-PD value in POST COVID-19 eyes compared to the healthy group (p<0.04) has been observed.
Once again, we thank the reviewers for their precious inputs and the time they put in reviewing our paper and look forward to meeting their expectations. We hope that these changes meet your requirements.
The authors
Reviewer 2 Report
I contacted the editor regarding this manuscript. Although the paper seems well-written and is of high interest during this awful pandemic, the downloadable manuscript that was available to review DID NOT include any figures/graphs or tables.
My only substantive comment is with respect to Figure 2. The figure is well-designed, but ALL of the labels throughout the figure are WAY TOO SMALL for interpretation by the reader. Please make a full-faith effort to improve the ease of viewing of Figure 2.
Author Response

(The authors gave the same response as above.)

Round 2
Reviewer 2 Report
the authors have adequately taken my comments/suggestions into consideration.